# Learning to Route Languages for Multilingual Policy Optimization

**Geyang Guo** [1]  **Hiromi Wakaki** [2]  **Yuki Mitsufuji** [2][3]  **Alan Ritter** [1]  **Wei Xu** [1]

## Abstract

Large language models (LLMs) are trained on heterogeneous multilingual corpora, yet existing policy optimization methods often implicitly restrict each training question to a single response language or rely on a fixed dominant language for supervision. We propose language-routed policy optimization (LRPO), an online reinforcement learning framework that treats language as a selectable variable. LRPO elicits multilingual rollouts for each training question and integrates their relative quality into preference-based policy updates, increasing the diversity and informativeness of training signals under the fixed rollout budget. To adaptively determine which languages to explore during reinforcement learning, we introduce a trainable language router formulated as a multi-armed bandit, balancing exploration of underutilized languages with exploitation of more informative ones. Extensive experiments show that LRPO consistently improves multilingual performance, demonstrating that adaptive language routing enables effective cross-lingual knowledge exploitation for training. We release all the resources at https://github.com/Guochry/LRPO.

## 1. Introduction

Trained on massive and heterogeneous corpora, large language models (LLMs) potentially integrate knowledge across diverse sources, domains, and languages. For example, LLMs can assist with literature search across languages and less accessible sources (Bubeck et al., 2025), enabling researchers to access and integrate a wider range of information. More broadly, information quality and coverage vary greatly across languages, suggesting that knowledge expressed in different languages can be complementary. If models can more effectively leverage such cross-lingual

[1]Georgia Institute of Technology [2]Sony Group Corporation [3]Sony AI. Correspondence to: Geyang Guo <guogeyang@gatech.edu>.

*Proceedings of the 43rd International Conference on Machine Learning*, Seoul, South Korea. PMLR 306, 2026. Copyright 2026 by the author(s).

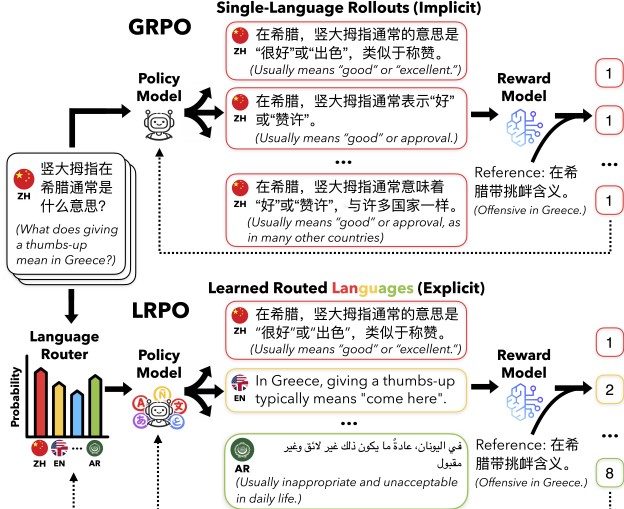

*Figure 1.* Comparison of GRPO and LRPO rollout process. While standard GRPO (top) is restricted to the source language, LRPO dynamically explores rollout languages during training. In this example, while Chinese and English responses provide common misconceptions about Greek etiquette, LRPO routes the query to an Arabic rollout that captures the correct cultural nuance, leading to higher reward and more accurate knowledge grounding.

knowledge, they may not only improve multilingual performance but also use strengths from one language to compensate for weaknesses in another. This raises a fundamental question: *How can LLMs better exploit knowledge distributed across languages and underrepresented sources?*

To elicit knowledge encoded during pre-training more effectively, recent work explores fine-tuning algorithms such as reinforcement learning from human feedback (RLHF) (Ouyang et al., 2022) and group relative policy optimization (GRPO) (Shao et al., 2024). These methods collect rollout generations from the current policy, evaluate their quality, and update the policy accordingly. However, such rollouts for each training question are often confined to one single given language, improving performance within that language while leaving cross-lingual generalization to implicit internal mechanisms.

To further improve performance across languages, recent policy optimization methods (She et al., 2024; Zhao et al., 2025b) explicitly model multilingual capabilities by assuming that a fixed dominant language (e.g., English) provides

more informative supervision than others (e.g., non-English). Dominant-language responses are therefore used as anchors for preference computation, which has been effective for tasks such as math reasoning and general safety. However, this assumption does not universally hold. Knowledge is unevenly distributed across languages, and its quality often depends on regional and contextual alignment (Guo et al., 2025b). Consequently, relying on a single dominant language, either for inference or training, can limit the model's ability to *explore* and *exploit* the most informative language, thereby overlooking underrepresented knowledge embedded in the model (see Figure 1 for an example).

In this work, we propose **language-routed policy optimization** (LRPO), an online reinforcement learning (RL) framework that integrates cross-language generations guided by a trainable language router during training. Instead of restricting each question to a single response language, LRPO elicits generations across languages, enabling the model to surface complementary knowledge and increase the diversity and informativeness of training signals.

To obtain reliable quality feedback for multilingual generation, LRPO evaluates each generation against high-quality references using cross-lingual semantic similarity (Chang et al., 2025). A key challenge is that raw similarity scores are not directly comparable across languages: semantically equivalent content may receive systematically different scores depending on the language pair, which can distort relative preferences within a multilingual rollout group. To address this, LRPO estimates cross-lingual similarity statistics offline and calibrates quality signals during training. This enables reliable cross-language comparison within each rollout group, allowing the model to favor informative content regardless of the language in which it is expressed.

To further determine which languages to explore for each question, we introduce a trainable language router that dynamically assigns preference over candidate languages during training. We formulate language selection as a multi-armed bandit problem, balancing exploration of underutilized languages with exploitation of languages that provide more informative training signals. Extensive experiments across three model families (i.e., Qwen, Llama, and Gemma) and five multilingual benchmarks demonstrate that LRPO consistently improves multilingual performance. For example, on Qwen2.5-1.5b, LRPO improves the average score on mGSM-v2 (Peter et al., 2025) from 24.87 to 38.25, and more broadly, improves seen-language average score across all five benchmarks, including CARE (Guo et al., 2025b), CARE-pro, mGSM-v2 (Peter et al., 2025), Global-MMLU-Lite (Singh et al., 2025), and Include-Lite (Romanou et al., 2024), by +5.08 and +2.85 points over the initial instruction-tuned model and the GRPO method.

## 2. Related Work

**Multilingual Performance Gaps in LLMs.** LLMs exhibit performance gaps across languages. On reasoning tasks, high-resource languages such as English often dominate, reflecting imbalances in training data and optimization phases (Kang et al., 2025; Zhao et al., 2025a). However, this pattern does not hold for factual questions (Wang et al., 2025a) or broader real-world user queries (Liu et al., 2025), where translation-based approaches may miss language-dependent variation in model reliability, especially for regional knowledge grounded in language and culture (Guo et al., 2025b). Moreover, stronger alignment between representations of parallel inputs across languages does not consistently yield improved performance (Ravisankar et al., 2025; Ai et al., 2025), suggesting that representational alignment alone is still insufficient. Together, these findings suggest that different languages encode complementary strengths, highlighting the need for optimization methods that explicitly adapt to differences across languages.

**Multilingual Policy Optimization.** Prior work explores various methods to improve multilingual performance, either by introducing cross-lingual chains of thought at inference time (Son et al., 2025; Zheng et al., 2025; Li et al., 2025), or by proposing multilingual training algorithms (She et al., 2024; Yang et al., 2024b; Zhao et al., 2025b; Hwang et al., 2025; Zhang et al., 2025b). However, these approaches largely rely on the shared assumption that high-resource languages, most notably English, provide higher-quality generations. Yet, this assumption does not universally hold, as prior work indicates that different task domains favor different languages (Huang et al., 2024), while leaving the interaction between such language advantages and a model's internal knowledge underexplored. Another line of work constructs multilingual preference training data through translation (Dang et al., 2024) or human annotation (Guo et al., 2025b; Wang et al., 2025b; Zhang et al., 2025a), but such data is costly to collect and scale, further motivating the need for more effective ways of expanding and utilizing existing data.

## 3. LRPO Method

In this section, we introduce LRPO, which includes a trainable *language router* that dynamically assigns rollout languages (§3.1), updates the policy using multilingual relative preference signals (§3.2), and jointly optimizes the language router to favor more informative languages (§3.3).

### 3.1. Language Router Design

For each training question $x$ written in input language $\ell_x$, the language router selects rollout languages that are most effective for policy optimization. We consider a language

**Algorithm 1** Language-routed multilingual rollout group

---

**Input:** Training question $x$ in language $\ell_x$, topic $t(x)$, optional region $g(x)$; topic matrix $\mathbf{A}$, region matrix $\mathbf{B}$; rollout size $K$; on-policy quota $K_{\mathrm{on}}$; policy $\pi_\theta$.
**Output:** Rollout group $\mathcal{G} = \{y_k\}_{k=1}^K$ in languages $\{\ell_k\}_{k=1}^K$.
1: $\mathbf{z} \leftarrow \mathbf{A}[t(x), :] + \mathbb{I}[g(x) \neq \varnothing]\, \mathbf{B}[g(x), :]$
2: $\mathbf{p} \leftarrow \mathrm{Softmax}(\mathbf{z}/\tau)$
3: $\ell_{1:K} \leftarrow [\underbrace{\ell_x, \ldots, \ell_x}_{K_{\mathrm{on}}},\ \mathrm{Sample}_\epsilon(\mathbf{p},\ K - K_{\mathrm{on}})]$
4: **for** $k = 1$ **to** $K$ **do**
5: $\quad y_k \sim \pi_\theta(\,\cdot \mid x,\ \ell_k\,)$
6: **end for**
7: **return** $\mathcal{G}$ and $\{\ell_k\}_{k=1}^K$

---

effective if it provides informative training signals and contributes meaningfully to improving the model's current capabilities. Accordingly, language selection should depend on both the content of the input question and the model's evolving multilingual capabilities. To capture content-level differences, we assign each training question to a high-level topic (e.g., reasoning, regional knowledge, etc.). Knowledge availability and quality vary substantially across languages for different topics, and model performance exhibits corresponding language-dependent variation. In particular, regional knowledge exhibits a distinctive structure: questions about a specific region are often best approached using languages spoken in or closely associated with that region, where native sources provide more accurate information.

To capture these patterns, we parameterize the language router with two matrices: a *topic-by-language* matrix $\mathbf{A}$ and a *region-by-language* matrix $\mathbf{B}$. For a training question $x$ with topic $t(x)$, the router first retrieves the corresponding topic-level logits $\mathbf{A}_{t(x)}$. If the question is associated with a region $g(x)$, the router additionally incorporates region-level logits $\mathbf{B}_{g(x)}$. The combined logits are converted into a language distribution using a temperature-scaled softmax:

$$p(\ell \mid x) \propto \exp\left(\frac{A_{t(x),\ell} + \mathbb{I}[g(x) \neq \varnothing]\, B_{g(x),\ell}}{\tau}\right), \quad (1)$$

where $\tau$ controls the smoothness of the distribution and $\mathbb{I}[\cdot]$ is the indicator function.

Based on the resulting distribution, the router samples languages to construct a multilingual rollout group of size $K$. To preserve on-policy learning for the input language, we reserve a fixed quota of $K_{\mathrm{on}}$ rollouts generated in the language of the original question, and sample the remaining $K - K_{\mathrm{on}}$ rollouts according to the router-determined distribution. The policy model follows the sampled language, via language tags or target-language system prompts, to generate rollouts accordingly; implementation details are provided in §5.4. The overall procedure is summarized in Algorithm 1.

## 3.2. Policy Update with Language-Routed Rollouts

Given the multilingual rollout group in §3.1, policy optimization then leverages these cross-lingual generations to improve model behavior by exploiting complementary knowledge across languages. Each of the $K$ rollouts is evaluated along two dimensions: *response quality* and *language consistency*.

**Quality Reward and Cross-lingual Calibration.** To obtain reliable quality feedback $r^{\mathrm{qual}}$ for multilingual generations, LRPO evaluates each rollout response against a high-quality reference response using semantic similarity (Chang et al., 2025). A key challenge in the multilingual setting is that raw similarity scores are not directly comparable across languages, which can bias relative preferences within a multilingual rollout group (detailed analysis in §5.4). To mitigate this issue, LRPO adopts a two-stage cross-lingual reward calibration strategy, consisting of *offline estimation* and *online calibration*.

During offline estimation, for each unordered language pair $\langle \ell_i, \ell_j \rangle$, we collect response pairs that characterize the similarity distribution between a reference response $z^{(\ell_i)}$ and candidate responses in $\ell_j$. Specifically, we consider three types of pairs: (1) *semantically equivalent pairs* $(z^{(\ell_i)}, z^{(\ell_j)})$, where $z^{(\ell_j)}$ is a semantically equivalent expression in language $\ell_j$, representing upper-bound cross-lingual alignment; (2) *naturally mismatched pairs* $(z^{(\ell_i)}, z'^{(\ell_j)})$ with $z' \neq z$, representing semantically unaligned responses; and (3) *hard contrastive pairs*, defined as the most similar mismatched pairs. These scores form a language-pair-specific empirical distribution $\mathcal{S}_{\ell_i, \ell_j}$, which is then used to estimate calibration statistics.

During RL training, for each rollout response $y^{(\ell_j)}$ and reference response $z^{(\ell_i)}$, LRPO first computes the raw similarity score $s(y, z) = \mathrm{sim}(y^{(\ell_j)}, z^{(\ell_i)})$, then calibrates it into a quality reward $r^{\mathrm{qual}}(y)$ using one of two following methods. The first is a mean-based calibration, where $\mathcal{S}_{\ell_i, \ell_j}$ is summarized by the mean value $\mu_{\ell_i, \ell_j}$ across semantically equivalent pairs:

$$r^{\mathrm{qual}}(y) = s(y, z) - \lambda\big(\mu_{\ell_i, \ell_j} - \mu_{\mathrm{ref}}\big), \quad (2)$$

where $\mu_{ref}$ denotes a global reference mean across all language pairs, and $\lambda$ controls the calibration strength.

The second is a distributional quantile-based calibration, where we estimate the empirical quantile function $\mathcal{Q}_{\ell_i, \ell_j}$ from $\mathcal{S}_{\ell_i, \ell_j}$ and map the raw similarity score to its quantile:

$$r^{\mathrm{qual}}(y) = \mathcal{Q}_{\ell_i, \ell_j}\big(s(y, z)\big). \quad (3)$$

The calibrated reward is more comparable across languages and can be directly used within each multilingual rollout group, allowing the policy to favor more informative responses regardless of the language they are written in.

**Algorithm 2** Language-Routed Policy Optimization

---

**Input:** Training data $\mathcal{D}$; policy $\pi_\theta$; router parameters $\mathbf{A}, \mathbf{B}$.
1: **for** each training step **do**
2:     Sample a minibatch of questions $x \sim \mathcal{D}$
3:     **for** each $x$ **do**
4:         Sample multilingual rollout group $\mathcal{G}$ guided by the router following Algorithm 1
5:         Compute cross-lingual rewards $r_k$ in Equation 5 for each $y_k \in \mathcal{G}$
6:     **end for**
7:     Update policy $\pi_\theta$ with rewards $\{r_k\}$ with GRPO Objective
8:     **if** router update step **then**
9:         Aggregate rewards and compute $\bar{r}_{t,g,\ell}$ in Equation 6
10:        Update router parameters $\mathbf{A}, \mathbf{B}$ following Equation 7
11:    **end if**
12: **end for**

---

**Language Consistency Evaluation.** To ensure that the policy follows the routed language selection and thus exposes its knowledge encoded in diverse languages, we introduce a language consistency indicator that checks whether each response is generated in the target language:

$$r^{\text{lang}}(y_k) = \mathbb{I}[\text{Lang}(y_k) = \ell_k], \tag{4}$$

where $\text{Lang}(\cdot)$ denotes a language identification function.

**Policy Update.** A rollout contributes its quality reward only if it follows the routed language; otherwise, its reward is set to zero. Formally, the final reward is defined as:

$$r_k = r_k^{\text{qual}} \cdot r_k^{\text{lang}}. \tag{5}$$

We then apply GRPO (Shao et al., 2024) using the gated rewards $\{r_k\}$, where rewards are normalized within each multilingual rollout group.

### 3.3. Router Learning and Adaptation

Based on the policy's generations in various languages on each question, we update the router to **(1)** favor language channels that yield higher expected rewards for each topic or region, thereby eliciting higher-quality knowledge and making more efficient use of the rollout budget, and **(2)** maintain a balance between exploration and exploitation.

During policy optimization, we maintain a reward buffer that records rollout rewards indexed by topic, region, and language. We update the router every $M$ policy gradient steps to avoid overly sparse or noisy signals, by aggregating the rewards accumulated since the previous router update and computing the empirical mean for each $(t, g, \ell)$ tuple:

$$\bar{r}_{t,g,\ell} = \mathbb{E}[r_k \mid t(x) = t,\ g(x) = g,\ \ell_k = \ell], \tag{6}$$

which serves as an estimate of the expected utility of selecting language $\ell$ under the corresponding content condition.

Router learning can be viewed as a contextual multi-armed bandit problem, where each language corresponds to an action and the objective is to estimate its expected reward conditioned on topic and region. To prevent the router from collapsing to a small subset of languages and leaving multilingual channels underexplored, we adopt several mechanisms to balance exploration and exploitation. First, an $\epsilon$-greedy strategy assigns every language a non-zero probability of being sampled. Second, we apply simulated annealing to both the exploration rate $\epsilon$ and the softmax temperature $\tau$, encouraging exploration in early stages and shifting toward exploitation as training progresses.

Finally, we update router logits $\mathbf{A}$ and $\mathbf{B}$ using an exponential moving average with adaptation rate $\alpha$, yielding a stable online estimate of language utility for routing decisions, as follows:

$$\begin{aligned}
A_{t(x),\ell} &\leftarrow (1 - \alpha)\, A_{t(x),\ell} + \alpha\, \bar{r}_{t,g,\ell}, \\
B_{g(x),\ell} &\leftarrow (1 - \alpha)\, B_{g(x),\ell} + \alpha\, \bar{r}_{t,g,\ell}.
\end{aligned} \tag{7}$$

Formally, the overall training procedure is summarized in Algorithm 2, integrating language-routed rollouts, policy optimization, and router adaptation.

## 4. Experiment Setup

In this section, we describe the experimental details for evaluating the efficacy of LRPO in comparison to other training mechanisms across models, tasks, topics, and languages. The results and analyses are discussed in the next section.

### 4.1. Baseline Methods

We consider the following baselines for comparison: (1) DPO (Rafailov et al., 2023): an offline RL method that updates the policy by maximizing the log-odds difference between preference pairs. (2) MAPO (She et al., 2024): a multilingual policy optimization approach that treats English generations as higher-quality references and computes reward scores by measuring alignment (returned by a translation model) between multilingual candidate responses and English ones. (3) LIDR (Yang et al., 2024b): a multilingual policy optimization method that constructs same-language preference pairs by translating responses across languages, using translations of English ones as higher-quality anchors. (4) MPO (Zhao et al., 2025b): another multilingual policy optimization method that aligns log-odds differences between the preference pairs in other languages with those in the dominant language. (5) GRPO (Shao et al., 2024): an online RL algorithm that samples groups of responses, scores their quality, and updates the policy based on group-level feedback. We implement reward score by calculating similarity between the model's generations and reference responses (Chang et al., 2025) using mmBERT (Marone et al., 2025) as we use in LRPO. More details are in Appendix C.1.

| Approach | CARE | CARE-pro | | | mGSM-v2 | | | Global-MMLU-Lite | | | Include-Lite | | | Overall | | |
|---|---|---|---|---|---|---|---|---|---|---|---|---|---|---|---|---|
| | Avg. | Seen | Unseen | Avg. | Seen | Unseen | Avg. | Seen | Unseen | Avg. | Seen | Unseen | Avg. | Seen | Unseen | Avg. |
| *LLaMA3.2-1b-it* | | | | | | | | | | | | | | | | |
| Vanilla | 17.51 | 3.50 | 2.26 | 3.09 | 23.66 | 8.70 | 18.22 | 43.55 | 30.75 | 40.13 | 29.28 | 27.17 | 27.84 | 23.50 | 17.22 | 21.36 |
| DPO | 16.95 | 3.01 | 1.64 | 2.55 | 26.00 | 12.80 | 21.20 | 43.41 | 31.19 | 40.15 | 29.66 | 27.14 | 27.94 | 23.81 | 18.19 | 21.76 |
| MAPO | 16.84 | 3.50 | 2.46 | 3.15 | 25.09 | 11.20 | 20.04 | 43.68 | 30.56 | 40.18 | 29.57 | 27.22 | 27.97 | 23.74 | 17.86 | 21.64 |
| LIDR | 16.31 | 6.48 | 1.03 | 4.66 | 23.89 | 10.70 | 19.09 | 43.66 | 30.81 | 40.23 | 29.52 | 27.52 | 28.16 | 23.97 | 17.52 | 21.69 |
| MPO | 17.67 | 5.53 | 3.49 | 4.85 | 24.51 | 9.00 | 18.87 | 43.43 | 30.75 | 40.05 | 29.30 | 27.18 | 27.86 | 24.09 | 17.61 | 21.86 |
| GRPO | 20.15 | 2.91 | 3.33 | 3.05 | 24.91 | 14.80 | 21.24 | 44.45 | 31.44 | 40.98 | 29.59 | 27.03 | 27.84 | 24.00 | 18.40 | 22.18 |
| LRPO (Ours) | 18.62 | 3.96 | 3.08 | 3.66 | 25.31 | 12.20 | 20.55 | 43.80 | 31.19 | 40.43 | 29.67 | 27.47 | 28.17 | **24.27** | **18.48** | **22.29** |
| *Qwen2.5-1.5b-it* | | | | | | | | | | | | | | | | |
| Vanilla | 32.20 | 8.58 | 0.82 | 5.99 | 35.73 | 11.84 | 24.87 | 54.55 | 34.00 | 49.07 | 35.91 | 28.84 | 31.09 | 32.86 | 18.54 | 28.64 |
| DPO | 32.69 | 9.60 | 1.64 | 6.94 | 37.33 | 14.64 | 27.02 | 54.36 | 34.31 | 49.02 | 35.57 | 28.82 | 30.97 | 33.55 | 19.22 | 29.33 |
| MAPO | 31.98 | 6.20 | 0.62 | 4.34 | 36.80 | 12.24 | 25.64 | 54.55 | 34.06 | 49.08 | 35.37 | 28.92 | 30.97 | 32.47 | 18.52 | 28.40 |
| LIDR | 30.07 | 6.44 | 0.62 | 4.50 | 34.20 | 10.88 | 23.60 | 54.52 | 34.00 | 49.05 | 35.06 | 28.66 | 30.69 | 31.65 | 17.97 | 27.58 |
| MPO | 31.91 | 6.58 | 1.03 | 4.73 | 35.67 | 12.32 | 25.05 | 54.55 | 33.81 | 49.02 | 35.62 | 29.10 | 31.17 | 32.33 | 18.79 | 28.38 |
| GRPO | 33.66 | 8.26 | 0.82 | 5.78 | 43.43 | 12.90 | 32.33 | 54.09 | 34.88 | 48.97 | 36.00 | 29.18 | 31.35 | 35.09 | **19.44** | 30.42 |
| LRPO (Ours) | 34.60 | 12.36 | 1.23 | 8.65 | 52.51 | 13.30 | 38.25 | 53.05 | 32.81 | 47.65 | 37.16 | 28.90 | 31.52 | **37.94** | 19.08 | **32.15** |
| *Gemma3-4b-it* | | | | | | | | | | | | | | | | |
| Vanilla | 49.85 | 16.74 | 12.94 | 15.47 | 77.26 | 59.20 | 70.69 | 58.45 | 43.38 | 54.43 | 41.05 | 37.41 | 38.56 | 48.67 | 38.23 | 45.80 |
| DPO | 50.47 | 18.63 | 12.73 | 16.66 | 76.34 | 58.90 | 70.00 | 58.50 | 43.06 | 54.38 | 40.58 | 37.51 | 38.49 | 48.90 | 38.05 | 46.00 |
| MAPO | 50.22 | 17.69 | 11.50 | 15.62 | 76.67 | 60.20 | 70.68 | 58.30 | 43.25 | 54.28 | 40.94 | 37.35 | 38.49 | 48.76 | 38.08 | 45.86 |
| LIDR | 49.23 | 16.21 | 10.06 | 14.16 | 76.23 | 58.90 | 69.93 | 58.27 | 43.19 | 54.25 | 41.38 | 37.69 | 38.87 | 48.26 | 37.46 | 45.29 |
| MPO | 48.98 | 15.41 | 10.47 | 13.76 | 75.94 | 59.30 | 69.89 | 58.20 | 43.81 | 54.37 | 41.67 | 37.71 | 38.97 | 48.04 | 37.82 | 45.19 |
| GRPO | 51.24 | 19.75 | 17.25 | 18.92 | 78.00 | 59.20 | 71.16 | 57.34 | 41.25 | 53.05 | 41.88 | 37.62 | 38.98 | **49.64** | 38.83 | 46.67 |
| LRPO (Ours) | 51.26 | 19.26 | 16.84 | 18.45 | 77.49 | 62.10 | 71.89 | 58.45 | 43.31 | 54.42 | 41.10 | 37.18 | 38.43 | 49.51 | **39.86** | **46.89** |

*Table 1.* Overall performance on multilingual benchmarks across three model backbones. *"Seen"* denotes performance on languages included in the LRPO training process, while *"Unseen"* denotes performance on held-out languages not seen during training. CARE contains only seen languages by construction. The Overall score provides a general assessment across all benchmarks. CARE, CARE-pro, and mGSM-v2 are open-ended generation tasks; Global-MMLU-Lite and Include-Lite involve multiple-choice questions. Bold and underlined numbers denote the best and second-best results, respectively.

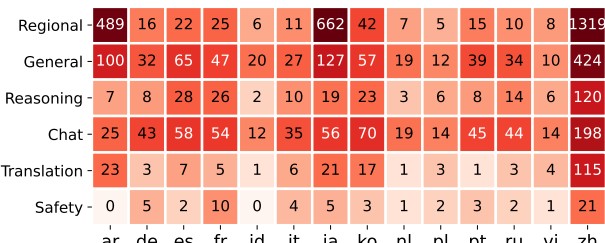

*Figure 2.* Distribution of topics across languages. Language labels follow ISO-639-1 codes; cell values denote raw example counts.

## 4.2. Implementation Details

**Model Details.** We evaluate LRPO on three instruction-tuned backbones, `gemma3-4b-it` (Team et al., 2025), `llama3.2-1b-it` (Grattafiori et al., 2024), and `qwen2.5-1.5b-it` (Yang et al., 2024a), to assess its effectiveness and generality across model families.

**Training Data.** We use the training split of two multilingual human preference datasts, HelpSteer3 (Wang et al., 2025b) and CARE (Guo et al., 2025b), totaling 4,885 samples across 14 languages. Both datasets provide annotations from native speakers and cover scenarios reflecting multilingual users' real-world needs, such as reasoning, safety, and regional queries. More details are in Appendix A.1.

**Languages and Topics.** The training data covers a diverse set of languages and topics. To characterize topic coverage, we identify six high-level categories through a coarse inspection of the data: *regional knowledge*, *general knowledge*, *chat*, *reasoning*, *safety*, and *translation*, which broadly cover real-world use cases. We then automatically classify each example with `gpt-oss-120b`, achieving 98% agreement with native annotators on 200 randomly selected examples. Interestingly, we observe that a non-trivial fraction (13.1%) of *regional knowledge* samples are expressed in non-local languages (see Figure 1 for an example), suggesting topical focus and surface language are not always aligned. We report statistics in Figure 2 and prompts in Appendix A.2.

## 4.3. Evaluation Tasks

We evaluate models on both general-purpose and region-specific multilingual benchmarks. Specifically, we use mGSM-v2[1] (Peter et al., 2025) for math reasoning, Global-MMLU-Lite (Singh et al., 2025) and Include-Lite (Romanou et al., 2024) for factual knowledge, and CARE's test set (Guo et al., 2025b) for regional knowledge and conversational settings. More details in Appendix C.2.

[1] mGSM-v2 is a corrected revision of mGSM (Shi et al., 2022), the multilingual extension of GSM8K (Cobbe et al., 2021), with revised translations and the same evaluation format.

| Language Mix | CARE | CARE-pro | | | mGSM-v2 | | | Global-MMLU-Lite | | | Include-Lite | | | Overall | | |
|---|---|---|---|---|---|---|---|---|---|---|---|---|---|---|---|---|
| | *Avg.* | *Seen* | *Unseen* | *Avg.* | *Seen* | *Unseen* | *Avg.* | *Seen* | *Unseen* | *Avg.* | *Seen* | *Unseen* | *Avg.* | *Seen* | *Unseen* | *Avg.* |
| LRPO | 34.60 | **12.36** | **1.23** | **8.65** | **52.51** | **13.30** | **38.25** | 53.05 | **32.81** | **47.65** | 37.16 | 28.90 | 31.52 | **37.94** | **19.08** | **32.15** |
| | | | | | | *Fixed-router LRPO variants* | | | | | | | | | | |
| Monolingual | 33.66 | 8.26 | 0.82 | 5.78 | 43.43 | 12.90 | 32.33 | **54.09** | **34.88** | **48.97** | 36.00 | **29.18** | 31.35 | 35.09 | **19.44** | 30.42 |
| Input-dominant | **35.40** | 10.96 | 1.03 | 7.65 | 50.17 | 11.90 | 36.25 | 53.59 | 32.63 | 48.00 | 37.43 | 28.91 | 31.62 | 37.51 | 18.62 | 31.78 |
| EN-dominant | 33.85 | 11.03 | **1.44** | 7.83 | 52.34 | 12.60 | 37.89 | 53.59 | 33.19 | 48.15 | **37.53** | 29.00 | **31.72** | 37.67 | 19.06 | 31.89 |
| Uniform | 33.95 | 10.33 | 1.03 | 7.23 | 50.74 | 11.90 | 36.62 | 52.98 | 32.63 | 47.55 | 37.13 | 28.81 | 31.46 | 37.03 | 18.59 | 31.36 |
| | | | | | | *Dynamic-router LRPO variants* | | | | | | | | | | |
| W.o. uniform init | **35.16** | 10.09 | 0.82 | 7.00 | 49.77 | 11.60 | 35.89 | 53.25 | 31.50 | 47.45 | **37.82** | **29.16** | **31.91** | 37.22 | 18.27 | 31.48 |
| W.o. annealing | 33.91 | 12.01 | 0.62 | 8.21 | 51.03 | 12.30 | 36.95 | **53.66** | 32.75 | **48.08** | 36.98 | 28.50 | 31.20 | 37.52 | 18.54 | 31.67 |

*Table 2.* Performance comparison of LRPO variants with fixed and dynamic language routers. The fixed-router variants compare fixed language-mixing strategies: *monolingual* uses only the input language; *input-dominant* uses 75% input-language and 25% English rollouts; *EN-dominant* uses 25% input-language and 75% English rollouts; and *uniform* uses 25% input-language rollouts with the remaining rollouts sampled uniformly across other languages. The dynamic-router variants ablate key components of router learning, including uniform initialization and temperature annealing. Overall, multilingual rollout strategies outperform monolingual training, while the full dynamic LRPO router achieves the strongest overall performance.

While these benchmarks cover a broad range of multilingual capabilities, two real-world multilingual information needs remain underrepresented: (1) fine-grained insider knowledge (e.g., city- or town-level facts), and (2) cross-lingual, cross-cultural information seeking. To address this gap, we introduce a new evaluation set, **CARE-pro**. It consists of two complementary components. The first focuses on fine-grained regional knowledge beyond country- or culture-level generalizations, with questions written by native annotators, targeting facts that are typically known only to locals (e.g. "*What is the small white room at Peking University used for?*"). The second captures cross-cultural curiosity, reflecting realistic information needs about foreign regions. Starting from multilingual multicultural QA datasets (Hasan et al., 2025; Chiu et al., 2024) authored by native speakers, we translate data samples into a different set of languages that are mother tongues of our own in-house annotators and ask annotators to assess cross-cultural relevance from a foreign perspective, filtering out samples that do not elicit genuine curiosity, and revising retained questions and answers when necessary. To ensure both quality and difficulty, we exclude samples that LLMs or search engines can readily answer. Annotators are instructed to produce concise, factual, and timeless questions with objective answers, and a second annotator validates each sample.

Following SimpleQA (Wei et al., 2024), we evaluate model outputs using a prompted LLM-based classifier (`gpt-oss-120b`), which compares the model response against the reference answer and assigns one of four labels: *correct*, *correct but wrong language*, *incorrect*, or *not attempted*. This setup achieves 93.5% agreement with human annotators. Additional details are in Appendix B.

## 5. Experiment Results

We compare LRPO with existing policy optimization methods (§5.1), analyze multilingual rollout groups (§5.2) and the dynamic router (§5.3), and finally examine the cross-lingual reward formulation (§5.4) and warm-starting (§5.5).

### 5.1. Main Results

Table 1 reports the performance of LRPO and baseline methods across multilingual benchmarks and model families. Our evaluation covers a broad set of languages, including benchmarks such as Include-Lite that span 44 languages. Since the evaluated languages include both those observed during LRPO training and held-out languages, we report results aggregated over *seen* and *unseen* language splits.

Across three model families, we observe that LRPO achieves the strongest overall performance. In most settings, LRPO leads to the best performance on seen languages while maintaining competitive results on unseen ones. For each benchmark, LRPO generally preserves overall performance on the short-form multiple-choice tasks, namely Global-MMLU-Lite and Include-Lite. More obvious gains are observed on CARE, CARE-pro, and mGSM-v2, all of which involve open-ended generation. CARE-pro in particular targets cross-lingual understanding, and the observed improvements suggest that dynamically selecting rollout languages provides more informative and complementary supervision signals. In the following sections, we conduct further analysis using `Qwen2.5-1.5b-it` to better understand LRPO's training process.

### 5.2. Analysis of Multilingual Rollout Ratio

To evaluate whether incorporating multilingual generations is beneficial and how different language compositions affect multilingual performance, we study LRPO with fixed, manually designed language logits for each topic and region. We evaluate four fixed language mixing strategies: (1) *monolingual*, where all rollouts use the input language; (2) *input-dominant*, with 75% of rollouts in the input language

**Question (JA)**
プラダの本店はミラノにありますが、フェラガモの本店はどの都市にありますか？ (*Translation:* Prada's flagship store is in Milan; which city is Ferragamo's flagship store in?)
**Ground truth:** フィレンツェ (*Translation:* Florence)

---

**Same-language rollout (JA, standard GRPO)**
フェラガモの本社はローマにあります。
(*Translation:* Ferragamo's headquarters are in Rome.)

- - - - - - - - - - - - - - - - - - - - - - - - -

**Cross-lingual rollout (EN, included in LRPO)**
Ferragamo's flagship store is located in Rome, Italy.

- - - - - - - - - - - - - - - - - - - - - - - - -

**Cross-lingual rollout (FR, included in LRPO)**
Le siège de Ferragamo se trouve à Florence.
(*Translation:* Ferragamo's headquarters are in Florence.)

*Table 3.* Qualitative case study of rollout language effects, that illustrates the limitations of same-language rollouts and cross-lingual exploration benefits. Gray denotes English translations (added for reader's convenience); green and red highlight correct and incorrect facts, respectively.

and 25% in English; (3) *English-dominant*, with the inverse ratio (i.e., 25% input, 75% English); and (4) *uniform*, where 25% of rollouts use the input language and the remainder are sampled uniformly across all supported languages.

Table 2 shows that incorporating multilingual rollouts consistently improves overall performance compared to monolingual training. Both input-dominant and English-dominant mixtures outperform the monolingual baseline across most benchmarks, indicating that exposing the policy to responses in multiple languages during optimization is beneficial even with fixed routing. Notably, English-dominant mixing yields the strongest overall results among the fixed-routing variants, particularly on reasoning-heavy benchmarks such as mGSM-v2. However, uniform multilingual sampling does not further improve performance and, in some cases, underperforms alternative strategies, highlighting that naively increasing language diversity is insufficient.

**Qualitative Case Study.** We further present qualitative cases where the policy generates rollouts in different languages for the same training question. As shown in Tables 3, multilingual rollouts increase both the diversity and the informativeness of the rollout group, helping explain the performance gains observed in our experiments. Besides, these cases suggest that there is no universally optimal rollout language. Instead, the most informative language depends on the content of the question, as well as the current knowledge distribution of the policy model. In the presented example in Tables 3, rollouts in both the original input language (Japanese) and a high-resource language (English) produce incorrect answers, while a French rollout recovers a higher-quality signal. This illustrates that simply relying on the input language or a dominant language is insufficient. Consequently, to effectively explore and exploit informative

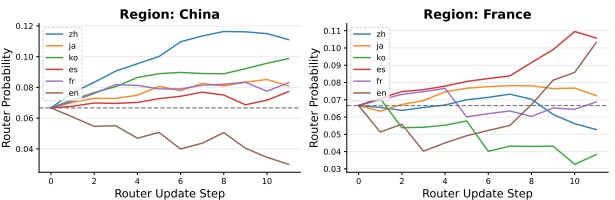

*(a)* Region-conditioned language probabilities over router updates, illustrated for *Chinese* and *French* regional questions.

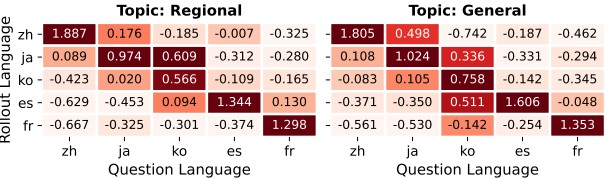

*(b)* Per-question advantages across languages (normalized calibrated rewards), on *regional* and *general* topics.

*Figure 3.* Language router behavior during optimization and cross-lingual quality differences. (**a**) Evolution of region-conditioned language probabilities over router updates. (**b**) Per-question advantages across languages, obtained by normalizing calibrated reward scores over all language responses to each question.

language channels during training, it is necessary to adopt a learned, context-dependent language routing strategy.

### 5.3. Analysis of Router Learning Dynamics

Compared with fixed, manually designed language routers, allowing the router to adapt during training generally leads to stronger performance. As shown in Table 2, dynamic routing outperforms LRPO with a fixed router, with the largest gain of $+1.33$ on seen languages in CARE-pro. Since CARE-pro evaluates fine-grained regional and cross-lingual understanding, this improvement suggests that dynamic language sampling can better align the model's multilingual knowledge with each training question.

**Router's Evolving Process.** Figure 3a illustrates how region-conditioned language probabilities evolve differently across regions. For questions related to China, the router increasingly favors Chinese, consistent with the strong Chinese capability of `Qwen`; for France, it shifts toward related languages such as Spanish. To further assess whether answering the *same* question in different languages leads to varying response quality, we conduct a controlled evaluation on both *regional* and *general* topics. We sample 20 questions per language per topic, and for each question, generate responses in all languages. We then compute per-question advantages by normalizing calibrated reward scores across all language responses to the same question. Figure 3b presents the resulting topic–language matrices, revealing clear disparities in response quality. Interestingly, languages associated with culturally related contexts (e.g., Chinese and Japanese) tend to exhibit similar relative advantages.

| Method | CARE | CARE-pro | | | mGSM-v2 | | | Global-MMLU-Lite | | | Include-Lite | | | Overall | | |
|---|---|---|---|---|---|---|---|---|---|---|---|---|---|---|---|---|
| | *Avg.* | *Seen* | *Unseen* | *Avg.* | *Seen* | *Unseen* | *Avg.* | *Seen* | *Unseen* | *Avg.* | *Seen* | *Unseen* | *Avg.* | *Seen* | *Unseen* | *Avg.* |
| Vanilla | 32.20 | 8.58 | 0.82 | 5.99 | 33.09 | 10.50 | 24.87 | **54.55** | 34.00 | **49.07** | 35.91 | 28.84 | 31.09 | 32.86 | 18.54 | 28.64 |
| Warm-start SFT | 33.05 | 9.66 | **1.44** | 6.92 | 43.43 | 9.70 | 31.16 | 54.16 | 34.44 | 48.90 | **38.06** | 29.07 | **31.93** | 35.67 | 18.66 | 30.39 |
| GRPO | 33.66 | 8.26 | 0.82 | 5.78 | 43.43 | 12.90 | 32.33 | 54.09 | 34.88 | 48.97 | 36.00 | **29.18** | 31.35 | 35.09 | **19.44** | 30.42 |
| LRPO | 34.60 | **12.36** | 1.23 | **8.65** | **52.51** | 13.30 | **38.25** | 53.05 | 32.81 | 47.65 | 37.16 | 28.90 | 31.52 | **37.94** | 19.08 | **32.15** |
| LRPO-Zero | 34.27 | 8.61 | 0.82 | 6.01 | 43.66 | **14.20** | 32.95 | 53.89 | 34.31 | 48.67 | 36.10 | 27.59 | 30.30 | 35.31 | 19.23 | 30.44 |
| LRPO-Fixed | **35.40** | 10.96 | 1.03 | 7.65 | 50.17 | 11.90 | 36.25 | 53.59 | 32.63 | 48.00 | 37.43 | 28.91 | 31.62 | 37.51 | 18.62 | 31.78 |
| LRPO-Fixed-Zero | 34.82 | 8.37 | **1.44** | 6.06 | 45.31 | 12.20 | 33.27 | 53.68 | **35.00** | 48.70 | 35.57 | 28.82 | 30.97 | 35.55 | 19.37 | 30.76 |

*Table 4.* Performance comparison under different initialization strategies. LRPO improves multilingual performance without warm-starting, whereas warm-start SFT further enhances results by stabilizing multilingual rollouts during training.

**Initialization and Annealing.** We implement router learning with uniform initialization over all languages and temperature-controlled sampling. Specifically, the router starts from a uniform distribution to avoid early bias, and sampling is controlled by a temperature initialized at 1.0 and exponentially annealed with a decay rate of 0.999 to a minimum of 0.3, gradually shifting from broad exploration to more focused language exploitation. Ablation results in Table 2 show that removing uniform initialization leads to a 0.67 points drop on average. Besides, disabling temperature annealing yields comparable overall performance but degrades performance on unseen splits, suggesting that early router diversity contributes to robustness.

## 5.4. Multilingual Reward Calibration

As discussed in §3.2, LRPO relies on similarity-based quality rewards that are calibrated to be comparable across languages. We construct three types of sample pairs for each unordered language pair $\langle \ell_i, \ell_j \rangle$ to estimate the cross-lingual similarity distribution offline. Specifically, *semantically equivalent pairs* are obtained by translating references using an open-source translation model (Aya-expanse-32b), with 30 pairs sampled per language pair. Figure 4 visualizes resulting similarity distributions, revealing cross-lingual variation even for semantically equivalent content. We additionally include *naturally mismatched pairs*, formed by randomly sampling 10 responses per reference, and *hard contrastive pairs*, obtained by selecting the top-2 most similar mismatched responses for each reference. Together, these samples form an empirical similarity distribution for each language pair; during training, the raw similarity score is calibrated using the precomputed statistics, reducing language-induced bias while preserving content-based comparisons. More discussions are included in Appendix D.

## 5.5. Language-Control Initialization

The policy models do not naturally follow the instruction to answer in the specified language, which may differ from the question language. We consider two approaches before LRPO training to elicit this behavior: **(1)** Warm-starting with supervised fine-tuning (SFT). Specifically, we use spe-

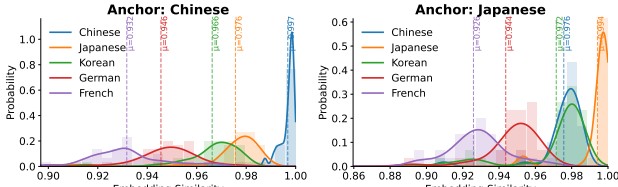

*Figure 4.* Uncalibrated cross-lingual similarity distributions. Each panel fixes an anchor language and shows similarity scores between reference responses and their semantically equivalent translations. Ideally, a well-calibrated similarity function should assign consistently high scores, close to 1.0, to equivalent responses across language pairs. However, despite controlling for semantic content, language pairs exhibit systematic shifts, indicating that raw cross-lingual similarity is not directly comparable.

cial language control tokens (e.g., `<|out_lang_zh|>`) to indicate the output language. We randomly sample 420 multilingual prompts from the Aya dataset (Singh et al., 2024), disjoint from our training data, and use `gpt-oss-20b` to generate two responses per prompt in two randomly selected target languages via language-conditioned prompts. We then format these cross-lingual question–answer pairs with language tags and apply SFT to initialize the policy. **(2)** Directly specifying the answering language in the system prompt without SFT, following DeepSeek-R1-Zero (Guo et al., 2025a). This is implemented by translating the base system prompt (i.e., "You are a helpful assistant.") into the target language and explicitly instructing the model to respond in that language.

From Table 4, we observe that LRPO outperforms GRPO even without warm-starting, indicating that cross-lingual exploitation can emerge naturally through reinforcement learning and benefit multilingual performance. Warm-starting with a lightweight SFT further improves performance by enhancing the model's ability to follow language tags, which stabilizes multilingual rollouts in the early training stages.

## 6. Conclusion

We present LRPO, a multilingual policy optimization framework that treats response language as an explicit training variable. By leveraging multilingual rollout groups, LRPO effectively explores knowledge across languages, increasing

the diversity and informativeness of training signals under a fixed rollout budget. Across three model backbones and five multilingual benchmarks, LRPO consistently improves performance; for example, on Qwen2.5-1.5b, LRPO achieves gains of $+5.08$ and $+2.85$ on seen languages over the initial instruction-tuned model and the GRPO method. Further analysis reveals that different languages contribute complementary strengths, highlighting the importance of explicitly modeling language-dependent variation in training.

## Impact Statement

The impact of this paper lies in introducing a training paradigm that improves multilingual performance by selectively leveraging knowledge expressed in different languages. By explicitly modeling topic- and region-dependent language strengths, LRPO moves beyond English-centric optimization and adapts the model's internal use of multilingual knowledge during training. This design enables the optimization process to better reflect language-dependent performance variation, leading to more accurate and contextually appropriate responses, particularly for queries involving regional knowledge or language-specific information. A broader impact of this work is its potential contribution to more inclusive and equitable multilingual AI systems. Many existing alignment and policy optimization pipelines implicitly prioritize dominant languages, which limits model effectiveness for non-English speakers and underrepresented regions. By incorporating multilingual rollouts and adaptive language routing, our approach demonstrates how knowledge distributed across languages can be more effectively utilized, helping reduce performance gaps and improve access to high-quality assistance for users interacting with models in their native languages. These benefits are especially relevant for applications such as multilingual information access, education, and cross-cultural communication.

In this study, we employed in-house annotators to collect regional question–answer data for CARE-pro. All annotators were university-level students fluent in the respective languages. Participation was voluntary, and no personal or sensitive information was collected from the annotators. Annotators were informed that the collected data would be used for the LLMs multilingual performance evaluation.

Overall, we believe there are many potential positive social benefits of our work. All datasets used in our experiments are publicly available, and we followed their respective licenses when conducting experiments.

## Acknowledgements

The authors would like to thank Nair Anupama, Navya Gautam, Rijul Magu, Xintong Qu, Kushal Ramaiya, Sanchez Vargas Carlos, Yiren Wang, and Dinesh Yadav for their assistance with data annotation. We would also like to thank Microsoft's Azure Accelerate Foundation Models Research Program and NVIDIA's Academic Grant Program for providing computational resources to support this work. This work was funded in part through a Sony Faculty Innovation Award, and by the NSF under grant number IIS-2144493, IIS-2052498 and SMA-2418946. Any opinions, findings, and conclusions or recommendations expressed in this material are those of the author(s) and do not necessarily reflect the views of the National Science Foundation.

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

| Language | Code | Alphabet | #Number |
|---|---|---|---|
| Arabic | ar | Arabic | 643 |
| Chinese | zh | Chinese | 2197 |
| Dutch | nl | Latin | 50 |
| French | fr | Latin | 167 |
| German | de | Latin | 107 |
| Indonesian | in | Latin | 41 |
| Italian | it | Latin | 93 |
| Japanese | ja | Kanji + Kana | 890 |
| Korean | ko | Hangul | 212 |
| Polish | pl | Latin | 42 |
| Portuguese | pt | Latin | 111 |
| Russian | ru | Cyrillic | 107 |
| Spanish | es | Latin | 182 |
| Vietnamese | vi | Latin | 43 |
| **Total** | | | **4885** |

*Table 5.* Languages in training data, along with their language codes, writing systems, and the number of samples per language.

## A. Data Details

### A.1. Training Data

Table 5 summarizes the language composition of the training data, which covers 14 languages spanning multiple writing systems. This diversity enables us to study multilingual policy optimization under realistic and heterogeneous data conditions.

### A.2. Topic and Region Classification

To obtain topic and region labels for each training sample, we use two prompt-based classifiers implemented with `gpt-oss-120b`. The topic prompt assigns each sample to one of six high-level topic categories based on its content: *regional knowledge*, *general knowledge*, *chat*, *reasoning*, *safety*, and *translation*. The second prompt identifies the geographic region associated with the query, which is classified as regional knowledge regardless of the language it is written in. These automatically assigned labels are used for parameterizing the language router described in §3.1. The full prompts are provided in Figures 5 and 6.

## B. CARE-pro

### B.1. Annotation Details

CARE-pro consists of two complementary annotation tracks: fine-grained insider knowledge and cross-lingual, cross-cultural information seeking. For fine-grained insider knowledge, native annotators are instructed to write questions targeting city- or town-level facts that are typically known only to insider residents, avoiding country-level generalizations

or widely documented information. For cross-cultural information seeking, annotators consider questions originally authored for one region and assess whether they elicit genuine curiosity from a foreign perspective after translation into their native language. Samples that do not reflect realistic cross-cultural interest are filtered out, and retained questions and reference answers are revised when necessary to ensure clarity and correctness. Across both tracks, annotators are instructed to produce concise, factual, and timeless questions with objective answers. Each sample is further validated by a second annotator to ensure quality and consistency. Figures 8–10 presents the annotation guideline for fine-grained insider knowledge, and Figures 11–13 shows the guideline for cross-cultural information seeking.

### B.2. Statistics

CARE-pro consists of 775 human-written samples spanning three languages: Chinese (204), Hindi (487), and Spanish (84). Each language includes two complementary components: fine-grained local insider knowledge and cross-cultural information seeking. Across languages, the dataset comprises 313 local insider knowledge questions and 462 cross-cultural inquiries. This composition supports the evaluation of both localized expertise and cross-cultural information needs in realistic multilingual settings.

### B.3. Evaluation Setup

We follow the evaluation protocol of SimpleQA (Wei et al., 2024) and use a prompted LLM-based classifier implemented with `gpt-oss-120b`. Given a model-generated response and a reference answer, the classifier assigns one of four labels: *correct*, *correct but wrong language*, *incorrect*, or *not attempted*. The resulting labels are used to compute accuracy, where only responses labeled as *correct* are counted as correct. Responses labeled as *correct but wrong language* are treated as incorrect to enforce language adherence. The prompt used for CARE-pro evaluation is shown in Figure 7.

## C. Implementation Details

### C.1. Baseline Methods

All baseline methods are trained with full-parameter tuning to ensure a fair comparison. For DPO, we use the `llama-factory` framework (Zheng et al., 2024), with a batch size of 128 on 4–8 NVIDIA A40 GPUs, and tune the learning rate over $\{3 \times 10^{-7}, 5 \times 10^{-7}\}$. For GRPO, we implement the method using the VERL framework (Sheng et al., 2025), with a training batch size of 256, 4 training epochs, a learning rate of $1 \times 10^{-6}$, and a rollout group size of 8. For MAPO, we follow the original setup and use `NLLB-600M-distilled` to compute alignment scores

| Calibration | CARE | CARE-pro | | | mGSM | | | Global-MMLU-Lite | | | Include-Lite | | | Overall | | |
|---|---|---|---|---|---|---|---|---|---|---|---|---|---|---|---|---|
| | Avg. | Seen | Unseen | Avg. | Seen | Unseen | Avg. | Seen | Unseen | Avg. | Seen | Unseen | Avg. | Seen | Unseen | Avg. |
| Vanilla | 32.20 | 8.58 | 0.82 | 5.99 | 33.09 | 10.50 | 24.87 | 54.55 | 34.00 | 49.07 | 35.91 | 28.84 | 31.09 | 32.86 | 18.54 | 28.64 |
| Quantile | 35.15 | 11.66 | 1.03 | 8.12 | 50.80 | 12.40 | 36.84 | 53.48 | 33.19 | 48.07 | 37.13 | 28.79 | 31.44 | 37.64 | 18.85 | 31.92 |
| Mean | 34.60 | 12.36 | 1.23 | 8.65 | 52.51 | 13.30 | 38.25 | 53.05 | 32.81 | 47.65 | 37.16 | 28.90 | 31.52 | 37.94 | 19.08 | 32.15 |

*Table 6.* Performance comparison under different multilingual reward calibration strategies.

| Method | Rollout time / step | Training time / step |
|---|---|---|
| GRPO | 32.07s | 327.24s |
| LRPO | 33.14s | 538.57s |

*Table 7.* Computational cost comparison between GRPO and LRPO. Both methods use the same rollout group size.

as suggested in the original work (She et al., 2024). We then construct preference pairs accordingly, perform DPO-style tuning, and report the results of the first iteration. For LIDR, we follow their procedure and use the same base language model as the rollout model to construct preference pairs. The model is then fine-tuned using DPO on the constructed preference data (Yang et al., 2024b). For MPO, we follow the official implementation and training configuration provided in the original paper (Zhao et al., 2025b).

### C.2. Evaluation Tasks

We evaluate models on a diverse set of existing multilingual benchmarks covering reasoning, general factual knowledge, and region-specific knowledge. For mGSM-v2, we use the mGSM version 2 dataset (Peter et al., 2025) and adopt 0-shot prompting, reporting accuracy as the evaluation metric. For Global-MMLU-lite (Singh et al., 2025) and INCLUDE (Romanou et al., 2024), we follow the standard evaluation pipeline provided by `lm-eval-harness` (Gao et al., 2024) and report accuracy. For CARE's test set (Guo et al., 2025b), we follow the original evaluation protocol and use an LLM-as-judge setup to assess model responses on a scale of 1 to 10. For the results reported in this paper, we multiply the mean score across samples by 10 to ensure comparable scales with other benchmarks. For CARE-Pro, we adopt 0-shot prompting and report accuracy based on the LLM-based evaluation described in Appendix B.

## D. Additional Results

### D.1. Reward Design

As introduced in § 3.2, we design two calibration methods for cross-lingual reward estimation, and compare their performance in this section. Table 6 shows that both mean-based and quantile-based calibration improve over other baseline methods, with mean-based calibration already providing a strong calibration baseline. Table 8 further illustrates why calibration is necessary: raw semantic similarity

can favor responses written in the same language as the reference, even when they are less informative or less aligned with the expected answer than responses in other languages. By correcting such language-pair-specific score biases, our calibration methods provide more reliable quality estimates across languages and support fairer comparison within multilingual rollout groups.

We note that our calibration framework is agnostic to the choice of similarity function and can be combined with alternative reward models. We leave further exploration of calibration strategies and reward signals to future work.

### D.2. Computational Cost

LRPO uses the same number of rollouts as GRPO, so the additional computation mainly comes from language routing and router updates. We provide the computational cost of GRPO and LRPO under the same hardware setup with 8 NVIDIA A40 GPUs. As shown in Table 7, rollout time increases slightly, potentially because LRPO encourages rollouts in non-English languages, where BPE tokenization typically produces more tokens than English.

### D.3. Statistical Significance

To assess the robustness of the results in Table 1, we conduct a paired bootstrap test at the language level with 20,000 samples, consistent with our evaluation protocol. LRPO shows statistically significant improvements over GRPO on Qwen2.5-1.5b (*Seen*: $p = 0.007$, *Average*: $p = 0.021$) and Gemma3-4b (*Average*: $p = 0.046$). The gains on LLaMA3.2-1b are not statistically significant, possibly due to weaker pretrained multilingual capability, as also observed in prior work (Chen et al., 2026).

| Language | Response | English Translation | Raw | Mean | Quantile |
|---|---|---|---|---|---|
| **Question:** Bodo Schäfer가 쓴 저서 *Die Gesetze der Gewinner*을 요약해줘. *(Summarize Bodo Schäfer's book Die Gesetze der Gewinner: Erfolg und ein erfülltes Leben.)* 
 **Reference:** *(The book presents principles for success and fulfillment, including mindset, goal setting, financial freedom, personal growth, time management, relationships, and resilience . . . )* | | | | | |
| Korean | 성공과 만족, 완성된 삶을 창조하는 법을 다룬 다... 계획과 행동, 재무 관리, 관계, 건강, 생각과 가치 등을 설명한다. | It discusses how to build success, satisfaction, and a fulfilled life . . . It covers planning and action, financial management, relationships, health, thoughts, and values. | 0.937 | 0.905 | 0.054 |
| Chinese | 本书介绍赢家在事业与生活中的行为模式...强调学习与生活平衡。 | It describes winners' behavioral patterns in work and life. . . emphasizing learning and life balance. | 0.934 | 0.926 | 0.056 |
| **Question:** 워드프레스 2011 테마와 2012 테마의 마크업 구조에서 변경된 부분과 그 이유를 설명해 줄 수 있니? *(Can you explain what changed in the markup structure between the WordPress 2011 and 2012 themes, and why?)* 
 **Reference:** *(The 2012 theme introduced HTML5 semantic elements and responsive layout . . . improve accessibility, mobile usability, performance, and maintainability.)* | | | | | |
| Korean | wp_head(), wp_meta(), wp_footer() 등의 함수 변화에 초점을 맞추며 . . . global scope 때문에 템플릿 내부 호출이 불가능해졌다고 말한다. | It mostly changes in functions such as wp_head(), wp_meta(), and wp_footer() . . . after WordPress 3.7, template-level calls became impossible due to global scope. | 0.987 | 0.955 | 0.549 |
| Chinese | 主要包括布局和结构变化、响应式设计、代码结构等 . . . 2012主题更强调现代、响应式和模块化设计。 | It includes layout and structure, responsive design, code organization . . . the 2012 theme emphasizes more modern, responsive, and modular design. | 0.971 | 0.962 | 0.560 |

*Table 8.* Examples illustrating cross-lingual reward calibration. For each question, we show the reference answer, rollout responses in different languages, and English translations. *Raw* is the uncalibrated semantic similarity, while *Mean* and *Quantile* are the rewards after mean-based and quantile-based calibration, respectively. These examples show that raw similarity scores are not always comparable across languages, and calibration can correct language-pair-level scale differences.

---

### Topic Classification

```
You are a classification model that assigns EXACTLY ONE topic label to the given user query.

Choose one and only one label from this list:

* [[Regional Knowledge]] | Questions whose core content concerns a specific country/region, culture, local
customs, geography, holidays, policies, or community-unique practices.
  Examples: "Why do Japanese people bow?", "What is Ramadan?", "Laws about traffic lights in Germany"

* [[General Knowledge]] | Factual questions that do NOT depend on a specific region or culture.
  Examples: "What is photosynthesis?", "How does a jet engine work?"

* [[Chat / Conversational]] | Greetings, opinions, or casual conversation without a clear problem-solving goal.
  Examples: "How is your day?", "Tell me something interesting"

* [[Reasoning / Logic]] | Math, coding, puzzles, or queries requiring structured inference or problem solving.
  Examples: "Solve this equation...", "Write Python to parse JSON"

* [[Safety / Ethics]] | Harmful, dangerous, sensitive, medical, political, or morally-charged topics.
  Examples: "How to hack...", "Is this drug safe for me?"

* [[Translation]] | Requests about translation, word meaning, or language comparison.
  Examples: "Translate this...", "What does X mean in English?"

---

GLOBAL PRINCIPLES:
1. Classify based on intent and meaning only | NEVER based on the language of the query.
2. If multiple labels seem possible, choose the SINGLE MOST SPECIFIC label that fits the user's intent.

---

OUTPUT REQUIREMENTS:
• Output ONLY one label.
• Format MUST be exactly:
  [[<label>]]
• The label MUST be exactly one of the six options above.
• No explanation, no additional text.

---

EXAMPLE 1:
Input Query:
初めまして。田中と申します。

Output:
[[Chat / Conversational]]

---

EXAMPLE 2:
Input Query:
quelle est la hauteur de la tour Eiffel

Output:
[[General Knowledge]]

---

EXAMPLE 3:
Input Query:
小粉红是什么

Output:
[[Regional Knowledge]]

---

Input Query:
{question}

Output:
```

*Figure 5.* Prompt template for question topic classification, used to assign each input query to exactly one topic category.

---

### Region Classification

```
You are a classifier that determines which REGION or CULTURE a user query is about. Given a user query:
1. Identify the region(s) or culture(s) the query is primarily about.
2. The region(s) should reflect cultural or country-level knowledge required to answer the question correctly.

IMPORTANT CONSTRAINTS:
• The region must be at the CULTURE or COUNTRY level.
  - Good: China, Japan, Korea, Arab World, United States, France
  - Bad: Asia, Europe, East Asia, Global, Chinese-speaking regions

• Do NOT infer the region from the language of the query alone.
  - The language used to ask the question may differ from the culture it refers to.

• If the query involves MULTIPLE distinct countries or cultures,
  output EACH of them explicitly.

• Do NOT merge multiple regions into a single broad category.
  - For example, output [[France]] [[Germany]] instead of [[Europe]].

• Only output regions that are NECESSARY to answer the question.

OUTPUT FORMAT:
Output one or more region labels.
Each label must be wrapped in double brackets.
Separate multiple labels with a single space.

Examples:
[[China]]
[[Japan]]
[[France]] [[Germany]]

No explanation.
No reasoning.
No additional text.

---

EXAMPLE 1
Input Query:
"小粉红是什么？"

Output:
[[China]]

---

EXAMPLE 2
Input Query:
"¿Por qué a los japoneses les gusta comer natto?"

Output:
[[Japan]]

---

EXAMPLE 3
Input Query:
"フランス革命は何ですか？"

Output:
[[France]]

---

EXAMPLE 4
Input Query:
"How do wedding traditions differ between China and Japan?"

Output:
[[China]] [[Japan]]

---

Input Query:
{question}
```

*Figure 6.* Prompt template for question region classification, used to assign each regional query to its related regions.

---

**CARE-pro Evaluation**

```
Your job is to look at a question, a gold target, and a predicted answer, and then assign a grade of either: ["CORRECT",
"CORRECT_BUT_WRONG_LANGUAGE", "INCORRECT", "NOT_ATTEMPTED"].
First, I will give examples of each grade, and then you will grade a new example.

The following are examples of CORRECT predicted answers.
Question: 阿曼允许在公共场合饮酒吗?
Gold target: 在公共场所饮酒或醉酒是法律禁止的行为。
Predicted answer 1: 不允许
Predicted answer 2: 在公共场合饮酒属于违法行为。

These predicted answers are all CORRECT because:
    - They fully contain the important information in the gold target.
    - They do not contain any information that contradicts the gold target.
    - Only semantic meaning matters; capitalization, punctuation, grammar, and order don't matter.
    - Hedging and guessing are permissible, provided the gold target is fully included and the response contains no incorrect
    information.
    - The predicted answer is in the **same language as the question**.

The following are examples of CORRECT_BUT_WRONG_LANGUAGE predicted answers.
Question: 谁发明了电话?
Gold target: 亚历山大·格拉汉姆·贝尔
Predicted answer 1: アレクサンダベル
Predicted answer 2: Invented by Alexander Bell.

These predicted answers are CORRECT_BUT_WRONG_LANGUAGE because:
    - The factual content is **correct** (same entity as the gold target).
    - But the predicted answer is **not in the same language as the question**.
    - Minor code-switching for proper nouns is allowed, but the overall answer here is clearly in a different language than
    the question.

The following are examples of INCORRECT predicted answers.
Question: 印度的 8 月 15 日是如何庆祝的?
Gold target: 这一天,印度各地都会举行升旗仪式、游行和文化活动来庆祝国庆节。
Predicted answer 1: 这一天主要是家人团聚并交换礼物。
Predicted answer 2: On this day, people light firecrackers and perform Lakshmi Puja, just like during Diwali.

Question: What are the names of Barack Obama's children?
Gold target: Malia and Sasha
Predicted answer 1: Malia.
Predicted answer 2: Malia, Sasha, and Susan.

These predicted answers are all INCORRECT because:
    - A factual statement in the answer contradicts the gold target.
    - Any incorrect factual content leads to INCORRECT | regardless of the language used.
    - Incorrect statements with hedging (\it is possible that...") are still considered incorrect.

The following are examples of NOT_ATTEMPTED predicted answers.
Question: 在印度文化中,新郎通常如何抵达婚礼现场?
Gold target: 新郎骑马,并伴随婚礼队伍一起到达。
Predicted answer 1: 这取决于家庭传统,我不敢给出确定的回答。
Predicted answer 2: 抱歉,没有足够信息回答这个问题。

Question: What are the names of Barack Obama's children?
Gold target: Malia and Sasha
Predicted answer 1: I don't know.
Predicted answer 2: Barack Obama has two children. I know that one of them is Malia, but I'm not sure about the other one.

These predicted answers are NOT_ATTEMPTED because:
    - The important information in the gold target is not included.
    - No statements in the answer contradict the gold target.
    - The answer expresses uncertainty without providing the correct entity.

Here is a new example. Simply reply with either CORRECT, INCORRECT, NOT ATTEMPTED. Don't apologize or correct yourself if
there was a mistake; we are just trying to grade the answer.
```
Question: {question}
Gold target: {target}
Predicted answer: {predicted_answer}
```

Grade the predicted answer of this new question as one of:
A. CORRECT
B. CORRECT_BUT_WRONG_LANGUAGE
C. INCORRECT
D. NOT_ATTEMPTED

Just return the letters "A", "B", "C", or "D", with no text around it.
```

*Figure 7.* LLM-as-a-judge prompt template for CARE-pro.

# Regional Knowledge QA Annotation Guideline

This task aims to collect and verify **regional knowledge questions** — concise, factual questions that reflect insider, place-specific knowledge about your city, town, or region. Each annotator should focus on their own familiar region.

## What Is a "Regional Knowledge Question"?

## Definition & Characteristics

- A *regional knowledge question* is one that only **local residents** typically know the correct answer to.
- It captures **local culture, habits, language, or traditions** that are not easily accessible to outsiders.
- At least in **one language (native or English)**, and at least one of Google search or current LLMs returns **unsatisfactory results**.
- Questions should be **specific, factual, and concise (1–2 sentences)**.
- Has a **single, factual, and timeless** answer (not opinion-based, not time-sensitive).

## Typical Types

Regional knowledge questions can cover a wide range of local aspects, including but not limited to:

1. **Language & Dialect** – local words, slang, or expressions
2. **Education & Institutions** – famous schools, entrance exams, or study habits
3. **Food & Dining** – local dishes, restaurants, eating customs, or etiquette
4. **Lifestyle & Daily Habits** – routines, markets, shopping, and social norms
5. **Events & History** – local festivals, traditions, or historical background
6. **Transportation & Geography** – routes, areas, or neighborhood nicknames
7. **Entertainment & Pop Culture** – local music, TV shows, celebrities, or sports teams
8. **Family & Relationships** – kinship terms, family habits, or social expectations
9. **Visiting & Greeting Etiquette** – how people welcome guests, polite phrases, local gestures
10. **Gift & Celebration Customs** – what gifts are appropriate, lucky/unlucky numbers, holiday practices
11. **Travel Recommendations** – local places to visit, insider-only experiences
12. **Religion & Beliefs** – local temples, holidays, or community beliefs (non-sensitive)
13. **Animals & Pets** – local pet culture, superstitions, or animal-related customs
14. **Clothing & Appearance** – weather-specific fashion, local clothing preferences
15. **Health & Wellness** – local remedies, spa culture, seasonal health tips

*Figure 8.* Annotation Guideline (1/6).

## How to Find Good Questions

Use two complementary approaches:

1. **Search-Based Discovery**

   - Explore regional media or online platforms such as *Rednote, Zhihu, Reddit, Naver Blog, YouTube, or local forums.*
   - Try search phrases like:
     - "Only locals in XX know"
     - "First time in XX"
     - "Questions about XX"
     - "Regional knowledge in XX"
   - Collect interesting questions that reveal *local-insider knowledge.*

2. **Brainstorming**

   - Reflect on your own experiences living there. Think of things outsiders often misunderstand or ask about.

🟨 *We are especially interested in questions that are hard to answer online, where, at least in one language (native or English), Google or other search engines return poor or unsatisfactory results.*

---

## Task Steps:

1. Write Questions (native language + English)
2. Write Correct Answers (native language + English)
3. Search and Record Results

- Search your question in the **native language** and **English**.
- Copy the main search results, or write **"NA"** if no relevant results appear.

1. Evaluate Quality

Mark for each language:

- ✅ **good** – search results are accurate or mostly correct
- ❌ **bad** – search results are wrong, missing, or unrelated

1. Test with an LLM

- Choose **one LLM** (e.g., GPT-4/5, Gemini, Claude, Qwen, or Llama, etc).
- Ask the same question (in your native language or English).
- Evaluate and record its quality as:

*Figure 9.* Annotation Guideline (2/6).

○ ✅ **good** – accurate or mostly correct

○ ❌ **bad** – wrong or misleading

## 💡 Tips

- Prefer questions that are *too local for Google or ChatGPT to answer perfectly*.
- Avoid sensitive political or personal content.
- Translation doesn't need to be word-for-word, but must preserve meaning.
- Even "bad" or "NA" results are valuable — please record them clearly.

*Figure 10.* Annotation Guideline (3/6).

# Cross-Cultural Curiosity QA Annotation Guideline

## Goal

This task explores how people from foreign cultures perceive foreign culture *native-written questions*. Each question has been automatically translated into **English** and your **native language**.

Your job is to:

1. Identify which questions are **culturally meaningful** — e.g., reflect Arabic regional knowledge or spark genuine curiosity.
2. For those meaningful questions, check **translation quality** and evaluate **how well Google and LLMs handle them**.

## Overall Workflow

For each row in the spreadsheet:

1. **Judge cultural meaningfulness**

   → Fill `Culturally Meaningful` as **Yes** or **No**.

2. **If "Yes"**:

   a. Check whether the **native-language translation** is natural and faithful.

   b. If not, rewrite under `Rewritten Question / Answer (native)`.

3. **For these processed QAs** (all labeled 1):

   ○ Use both the **English** and **native-language** versions to

   (a) search on Google and

   (b) ask an LLM.

   ○ Record **good** or **bad** for each.

## Step-by-Step Instructions

### Step 1 — Culturally Meaningful Judgment

**Column:** `Culturally Meaningful`

Mark **1** if the question is *culturally meaningful*, **0** otherwise.

**Definition — "Culturally Meaningful"**

*Figure 11.* Annotation Guideline (4/6).

- The question is about **the regional or cultural knowledge of that given `region`** — e.g., daily habits, local words, customs, norms, traditions, etc.
- It feels **interesting or worth learning** from your cultural perspective.
- The question should be specific, not overly broad or open-ended.

**Examples**

| Question (en) | Meaningful? | Reason |
|---|---|---|
| Why do Arab homes have courtyards? | yes | Reflects architecture & lifestyle |
| What is the capital of Saudi Arabia? | no | Generic fact |
| Who won the 2022 World Cup? | no | Not culturally unique |

## Step 2 — Translation Check & Rewriting

**Columns:** `Rewritten Question (native)`, `Rewritten Answer (native)`

If the native-language translation is:

- awkward, unnatural, or
- incorrect in meaning,

  then rewrite it to make it fluent and accurate while preserving the same sense.

**Definition — "Reasonable Translation"**

- Preserves **the same factual and cultural meaning** as the English version.
- Sounds **natural** to a native speaker (no literal machine-translation tone).
- Avoids distortion or missing context.

## Step 3 — Google Search Evaluation

**Columns:** `Search (en)` and `Search (native)`

Search each question on Google.

- For `Search (en)` : use the English version.
- For `Search (native)` : use the rewritten or original native-language version.

**Label:**

*Figure 12.* Annotation Guideline (5/6).

| Value | Meaning |
|-------|---------|
| **good** | Top 5 search results contain correct or mostly correct answers. |
| **bad** | Results are wrong, irrelevant, incomplete, or none appear. |

## Step 4 — LLM Evaluation

**Columns:** `LLM (en)`, `LLM (native)`, `LLM you choose`

Ask both the English and native-language versions to **the same LLM** (e.g., GPT-4, GPT-5, Gemini, Claude, Qwen, Llama).

**Label:**

| Value | Meaning |
|-------|---------|
| **good** | Answer is accurate or mostly correct. |
| **bad** | Answer is wrong, misleading, or unrelated. |

## 🪄 Additional Notes

- For all questions marked **"yes"** (culturally meaningful), complete the full evaluation steps.
- Ignore sensitive or opinion-based topics — just judge factual and cultural relevance.
- When unsure, mark "**no**" rather than guessing.

*Figure 13.* Annotation Guideline (6/6).

