# OpenReview forum: "Learning to Route Languages for Multilingual Policy Optimization"
_ICML.cc/2026/Conference — ICML 2026 regular_

### Official Review · Reviewer_fdaR · 2026-02-14

**Soundness:** 3
**Presentation:** 3
**Significance:** 2
**Originality:** 3
**Overall Recommendation:** 5
**Confidence:** 4

**Summary:**

This paper proposes LRPO (language routing policy optimization), which is a method for dynamically selecting straining languages during multilingual post-training. Instead of sampling languages uniformly or proportionally, the authors propose language selection as a multi-armed bandit problem, where each language represents an arm. They evaluate across multiple multilingual benchmarks and results show improvements over baseline sampling strategies.

**Compliance With Llm Reviewing Policy:**

Affirmed.

**Final Justification:**

Having reviewed the authors' response, I acknowledge that the authors have provided clarifications on the experiment setup and preliminary results or promised revisions to the suggested experiments. I have updated my score to 5 (accept).

**Key Questions For Authors:**

- Were there any observable patterns in the routing behavior that favor languages within the same language family? For example, if a query concerns Chinese culture, does the routing policy tend to prioritize languages such as Chinese, Japanese, or Korean? Perhaps an analysis grouped by language families or writing scripts could provide better insight into the trends observed in Figure 3 and help determine whether routing reflects shared cultural context.
- More qualitative analysis of the routing dynamics would strengthen the paper. For example, do certain languages consistently dominate for specific types of queries? Does the exploration actually benefit low-resource languages, particularly given that the EN-dominant variant achieves the highest overall score among the compared variants?
- How does the method scale with respect to the number of languages involved? What are the differences involving smaller sets of languages vs. larger sets of languages? And would the exploration cost grow linearly with the number of arms?

**Limitations:**

Not specifically about limitations but only the potential impact through the impact statement. Please refer to the weaknesses section for limitations identified by the reviewer.

**Strengths And Weaknesses:**

**Strengths**
- It is a well-motivated problem and the idea of framing language selection as a multi-armed bandit problem is interesting. It also seems to have positive downstream effects in terms of balancing exploration of under-utilized languages with exploitation of languages that yield stronger training signals.
- The routing mechanism is clearly described in the paper and grounded with established bandit optimization frameworks. It makes the suggested approach to be more modular.

**Weaknesses**
- There’s limited details on the human annotation procedures that were involved during the topic classification process of examples. How many annotators were involved and how the inter-annotator agreement was measured?
- Also, what is the language identification tool used to assign language labels in the training data, and what was the accuracy of this identification?
- In Table 1, the improvements in LRPO over baselines appear relatively small. Especially since the overall score aggregates across five distinct evaluation tasks, it’s unclear whether this improvement represents a superior performance of LRPO. Is there any statistical significance testing done? If so, the authors should note the standard deviations, p-vals and testing method.

---

> ### Author Rebuttal · Authors · 2026-03-31
>
> Thanks for your thorough review! We have addressed your comments below:
>
> >**There’s limited details on the human annotation procedures that were involved during the topic classification process of examples.**
>
> We use an open-source LLM rather than a human for topic classification (§4.2). To evaluate its reliability, we randomly sampled 200 examples and had one human annotator label them, observing a 98% agreement rate.
>
> >**What is the language identification tool used to assign language labels in the training data, and what was the accuracy of this identification?**
>
> We use FastText for language identification [1], which achieves over 95% accuracy on the WiLI-2018 benchmark and is widely adopted in prior work (e.g., NVIDIA NeMo-Curator). We will add these details to §4.2.
>
> >**In Table 1, the improvements in LRPO over baselines appear relatively small.**
>
> We conduct paired bootstrap testing at the language level, consistent with our evaluation protocol: for each of 20,000 samples, we resample languages within each benchmark, compute language-averaged scores, and aggregate across tasks to obtain p-values. We find that LRPO achieves statistically significant improvements over GRPO on **Qwen (Seen: p=0.007, Avg: p=0.021)** and **Gemma (Avg: p=0.046)**, while gains on LLaMA are not significant, possibly reflecting differences in base multilingual capability.
>
> >**Were there any observable patterns in the routing behavior that favor languages within the same language family?**
>
> This is a valuable suggestion. We will extend the analysis in Figure 3 by grouping languages into families for potential trends. Preliminarily, for regional questions in Korean, Japanese, or Chinese, rollout responses in this language family tend to receive higher reward scores compared to other languages.
>
> >**More qualitative analysis of the routing dynamics would strengthen the paper.**
>
> We conduct additional qualitative analysis by comparing an EN-dominant variant with exploration-enabled LRPO. Specifically on mGSM, while the EN-dominant variant achieves a better overall average score (+0.11), exploration improves performance on low-resource languages (bn, sw, te, th). Specifically, LRPO yields gains of +1.6 (sw), +1.2 (te), and +0.8 (th) over the EN-dominant variant. We will include more discussions on this in the paper.
>
> >**How does the method scale with respect to the number of languages involved? What are the differences involving smaller sets of languages vs. larger sets of languages? And would the exploration cost grow linearly with the number of arms?**
>
> We thank the reviewer for these interesting questions. We will add experiments in the revision to analyze how performance varies with the number of languages and how exploration cost scales with the number of arms. As a preliminary analysis on Qwen, we find that LRPO incurs a longer response compared with GRPO since LRPO encourages rollouts in non-English languages, where BPE tokenization typically produces more tokens. We will include a more detailed analysis in the final version.
>
> [1] A. Joulin, E. Grave, P. Bojanowski, M. Douze, H. Jégou, T. Mikolov. (2016). FastText.zip: Compressing text classification models.

---

> > ### Author Rebuttal · Reviewer_fdaR · 2026-03-31
> >
> > Thank you for the response. I acknowledge that the authors have provided clarifications on the experiment setup and preliminary results or promised revisions to the suggested experiments. I have updated my score accordingly.

---

> > > ### Author Response · Authors · 2026-04-04
> > >
> > > Thank you again for the thoughtful review and for raising your score. We are glad that the preliminary results and clarification addressed your concerns. As we mentioned in the rebuttal, we will incorporate the updated discussion in the future version of the paper.

---

### Official Review · Reviewer_GaXx · 2026-03-07

**Soundness:** 3
**Presentation:** 3
**Significance:** 2
**Originality:** 2
**Overall Recommendation:** 4
**Confidence:** 4

**Summary:**

This paper proposes an online preference optimization framework named LRPO. The proposed LRPO formulates the response language as a selectable variable rather than a fixed input constraint , based on which LRPO assigns a language router formulated as a multi-armed bandit to dynamically select rollout languages for each training question. Experiments on five real-world multilingual benchmarks across three model backbones demonstrate that LRPO improves multilingual performance.

**Compliance With Llm Reviewing Policy:**

Affirmed.

**Final Justification:**

The authors provided reasonable clarifications and supplementary evaluations (e.g., the Include-Lite results and calibration comparisons) that adequately addressed my initial technical concerns. While there is still room for future exploration, the rebuttal resolves my main reservations, so I have adjusted my score accordingly.

**Key Questions For Authors:**

Please refer to Strengths And Weaknesses

**Limitations:**

yes

**Strengths And Weaknesses:**

Strengths

1. This paper proposes a dynamic language routing scheme that treats language as an explicit variable, which balances the exploration of underutilized languages with the exploitation of informative ones without introducing an additional rollout budget.


2. The cross-lingual reward calibration strategy brings practical solutions to multilingual preference optimization, i.e., using offline similarity estimation to mitigate the bias in raw similarity scores across different language pairs.



Weaknesses

1. The authors may want to explain the motivation for relying on a massive external LLM (e.g., gpt-oss-120b) to assign discrete topic and region labels during training. The reliance on these external hard labels makes the pipeline heavy and limits the generalizability of the router.


2. The theoretical or empirical analysis of the phenomenon that cross-lingual similarity can be accurately calibrated by a single static pre-estimated mean ($\mu_{l_{i},l_{j}}$) is missing. The authors may want to consider how varying task complexities (e.g., reasoning vs. chat) might affect this similarity distribution.


3. The logical connection between the proposed language routing and the performance gains on short-form multiple-choice benchmarks (e.g., Global-MMLU-Lite and Include) is unclear. According to Table 1, the improvements on these benchmarks are marginal. Please reorganize the discussion to clarify whether LRPO mainly improves stylistic alignment rather than factual knowledge boundaries.


4. Some concerns about the implicit cross-lingual transfer mechanism are as follows.

(1) The authors may want to detail how optimizing a policy on a routed language (e.g., Arabic) implicitly improves the model's representation when queried in the source language (e.g., Chinese). The missing of the representation-level analysis makes the internal cross-lingual alignment mechanism unconvincing.

(2) I suggest the authors consider providing an end-to-end routing variant that directly extracts continuous routing features from the policy model's hidden states, rather than relying on prompt-based classifiers to determine the routing distribution.

---

> ### Author Rebuttal · Authors · 2026-03-31
>
> Thanks for your thorough review! We have addressed your comments below:
>
> >**W1: topic and region labels**
>
> It’s a straightforward six-way text classification task, so we use open-source LLM prompting (gpt-oss-120b) for simplicity and reproducibility. A smaller model, either fine-tuned or distilled from a larger model, would likely achieve comparable results.
>
> >**W2: cross-lingual similarity calibration for reward calculation**
>
> We considered the task complexities in the routing stage, as we needed rollouts to be in different languages. We calculate similarity between reference and LLM rollouts within each rollout group for the same question (which is for the same topic/task) to estimate relative quality, so we perform calibration across languages, not tasks.
>
> Cross-lingual similarity can be calibrated using a single pre-estimated global mean, not perfectly, but sufficiently for LRPO to work effectively to outperform GRPO (see Table 1). Inspired by your question, we added a new analysis by evaluating different reward models on held-out preference pairs from CARE and HelpSteer3 (excluding reference responses). The results show that mean calibration improves reward accuracy (84.44% vs. 82.78%) and better supports cross-lingual comparison within rollout groups.
>
> In addition, we explore an improved calibration that models the full similarity distribution for each language pair via quantile mapping. As shown in the table below, both mean calibration and quantile-based calibration achieve strong and comparable performance, consistently outperforming other baselines in Table 1. This suggests that mean calibration already serves as a strong and effective baseline for cross-lingual reward calibration in our setting, while more expressive calibration methods provide similar benefits.
>
> | Calibration strategy | CARE  | CARE (pro) | CARE (pro) | CARE (pro) | mGSM  | mGSM   | mGSM  | GMMLU-Lite | GMMLU-Lite | GMMLU-Lite | Include | Include | Include | Overall | Overall | Overall |
> | -------------------- | ----- | ---------- | ---------- | ---------- | ----- | ------ | ----- | ---------- | ---------- | ---------- | ------- | ------- | ------- | ------- | ------- | ------- |
> |                      | Seen  | Seen       | Unseen     | Avg.       | Seen  | Unseen | Avg.  | Seen       | Unseen     | Avg.       | Seen    | Unseen  | Avg.    | Seen    | Unseen  | Avg.    |
> | Mean                 | 34.60 | 12.36      | 1.23       | 8.65       | 52.51 | 13.30  | 38.25 | 53.05      | 32.81      | 47.65      | 50.13   | 35.43   | 40.11   | 40.53   | 20.69   | 33.85   |
> | Quantile             | 35.89 | 11.66      | 1.03       | 8.12       | 50.80 | 12.40  | 36.84 | 53.48      | 33.19      | 48.07      | 50.15   | 35.68   | 40.28   | 40.39   | 20.57   | 33.84   |
>
>
> >**W3: performances on factual knowledge benchmarks**
>
> We thank the reviewer for this insightful question. The marginal gains on Global-MMLU-Lite and Include are likely due to training data imbalance, as these benchmarks include more STEM questions with limited coverage in our data, and all methods show limited improvements. To validate this, we evaluate on Include-Lite (excluding STEM questions) and observe clearer gains across methods, with LRPO showing more pronounced improvements. This suggests LRPO improves not only stylistic alignment but also factual knowledge when the domain better aligns with training data. We will clarify this and include the analysis in the revision.
>
> | Qwen   | Vanilla | DPO   | MAPO  | LIDR  | MPO   | GRPO  | LRPO  |
> | ------ | ------- | ----- | ----- | ----- | ----- | ----- | ----- |
> | Seen   | 35.91   | 35.57 | 35.37 | 35.06 | 35.62 | 36.00 | 37.13 |
> | Unseen | 28.84   | 28.82 | 28.92 | 28.66 | 29.10 | 29.18 | 28.79 |
> | Avg.   | 31.09   | 30.97 | 30.97 | 30.69 | 31.17 | 31.35 | 31.44 |
>
> | Gemma  | Vanilla | DPO   | MAPO  | LIDR  | MPO   | GRPO  | LRPO  |
> | ------ | ------- | ----- | ----- | ----- | ----- | ----- | ----- |
> | Seen   | 41.05   | 40.58 | 40.94 | 41.38 | 41.67 | 42.04 | 40.46 |
> | Unseen | 37.41   | 37.51 | 37.35 | 37.69 | 37.71 | 37.65 | 37.41 |
> | Avg.   | 38.56   | 38.49 | 38.49 | 38.87 | 38.97 | 39.05 | 38.38 |
>
> | Llama  | Vanilla | DPO   | MAPO  | LIDR  | MPO   | GRPO  | LRPO  |
> | ------ | ------- | ----- | ----- | ----- | ----- | ----- | ----- |
> | Seen   | 29.28   | 29.66 | 29.57 | 29.52 | 29.30 | 29.59 | 29.67 |
> | Unseen | 27.17   | 27.14 | 27.22 | 27.52 | 27.18 | 27.03 | 27.47 |
> | Avg.   | 27.84   | 27.94 | 27.97 | 28.16 | 27.86 | 27.84 | 28.17 |
>
>
> >**W4: cross-lingual transfer mechanism**
>
> We focus on explicit multilingual rollouts in this paper to enable direct comparison with GRPO and other cross-lingual RL methods. Discussion of representation-level transfer is left for future work. However, we will be happy to add a comparison to the simple end-to-end routing baseline into the paper, which we expect to be slightly worse than the method we are using in the paper.

---

> > ### Author Rebuttal · Reviewer_GaXx · 2026-04-04
> >
> > Thank you for your detailed rebuttal. I will improve my score.

---

> > > ### Author Response · Authors · 2026-04-04
> > >
> > > Thank you again for the thoughtful review and for raising your score. We are glad that the additional experiments and clarification addressed your concerns. As we mentioned in the rebuttal, we will incorporate the updated discussion in the future version of the paper.

---

### Official Review · Reviewer_D52v · 2026-03-15

**Soundness:** 2
**Presentation:** 3
**Significance:** 2
**Originality:** 3
**Overall Recommendation:** 4
**Confidence:** 3

**Summary:**

The paper introduces Language-Routed Preference Optimization (LRPO), a framework for improving multilingual language model preference alignment by treating the response language as a selectable variable during preference optimization. Instead of generating rollouts only in the input language or relying on a fixed language, LRPO generates responses in multiple languages and learns to route training queries to the most informative languages using a trainable language router. To enable fair comparison of response quality across languages, a calibration procedure is introduced with cross-lingual similarity scores used as rewards. Experiments across multiple model families (Qwen, LLaMA, Gemma) and multilingual benchmarks show that LRPO improves multilingual performance compared to prior preference optimization methods such as GRPO and other multilingual preference training approaches.

**Compliance With Llm Reviewing Policy:**

Affirmed.

**Final Justification:**

I believe the rebuttals have addressed the main questions that I had through empirical experiments, and also, additional clarifications are helpful, which changed my evaluations, especially concerning whether the response language during test time would still be in the input language (at least the author seems to confirm this). The authors also provided some additional statistics that would be useful for future researchers to base their studies on.

**Key Questions For Authors:**

**Q1.** How would the proposed method be used in settings where there is constraint in the final answer language (e.g. through language forcing and also the language consistency being used)?

**Q2.** The training datasets used in the paper (HelpSteer3 and CARE) do not really have "golden" reference answers in the original dataset, so it would be helpful if the authors clarify how the reward signal is computed in the current setting and also when "gold" reference may not exist. Additionally, semantic similarity to references may bias optimization toward translation-like outputs or reference phrasing rather than overall response quality or reasoning correctness, particularly for open-ended tasks, so it would be nice if the authors can clarify regarding this.

**Q3.** Since most analysis are for Qwen, it would be nice to have additional analysis explaining why the gains differ across model families along with statistical significance testing or variance across runs.

**Limitations:**

See Weaknesses and Key Questions.

**Strengths And Weaknesses:**

## Strengths

**S1.** The paper introduces a novel training formulation by having a language router during online learning that allows the system to dynamically balance exploration and exploitation across languages depending on the question and model state.

**S2.** The experiments and analysis are relatively comprehensive by benchmarking several model families, multilingual preference training approaches, and on multiple multilingual benchmarks. Qualitative cases are also provided that demonstrate how generating responses in different languages can surface complementary knowledge and improve training signals.

## Weaknesses

**W1.** The practical usage of language routing remains somewhat unclear. While the authors do show increase in the performance, in many scenarios, users expect responses that are the same as the input language, so generating answers in different languages would be somewhat not desirable. In addition, if the model is weak in certain languages (such as low-resource languages), then this algorithm will probably rarely surface such languages and route to different language instead.

**W2.** The quality signal used during optimization is derived from cross-lingual semantic similarity with reference answers, so this may bias the optimization toward reference phrasing or translation-like outputs and may not fully capture response quality or reasoning correctness, particularly for open-ended generation tasks. In addition, the optimization relies on predefined topic and region signals, which require metadata annotations (as done by the authors using gpt-oss-120b); it is unclear how readily available such annotations would be in practical settings (or even with settings that are out of the predefined topic). Finally, the paper does not clearly discuss the computational overhead of LRPO or whether the rollout budget is comparable to baselines such as GRPO. It would also be useful to understand how performance changes under different rollout budgets.

**W3.** The performance looks strong on Qwen, but the rest of the model families (LLaMA and Gemma) look minimal given the complexity of the method. This raises questions about how consistently the approach benefits different architectures and whether the method scales effectively to larger models. Additionally, given the close difference in performance with other methods, the paper does not report statistical significance for the results, making it difficult to assess the robustness of the reported improvements.

---

> ### Author Rebuttal · Authors · 2026-03-31
>
> Thanks for your thorough review! We have addressed your comments below:
>
> >**W1/Q1: output language and handling of weaker (low-resource) languages**
>
> We will clarify this in the paper: (1) at test time, the model always responds in the input language; (2) only at training time, we use multilingual rollouts with a router to select and diversify signals (e.g., a Chinese query about Polish culture may benefit from rollouts in Polish and other European languages). Our LRPO is a train-time method.
>
> To better handle low-resource languages, we also explicitly incorporate exploration (e.g., epsilon-greedy and annealing) and reserve a fixed quota of rollouts in the input language (§3.3). This prevents collapse to dominant languages (e.g., English) and ensures that low-resource languages are sufficiently explored and improved during training. We will revise §3.1 to make these design choices and their practical implications clearer.
>
> >**W2: use of cross-lingual semantic similarity as rewards and computation time**
>
> We thank the reviewer for this insightful discussion. For general open-ended generation tasks, obtaining verifiable rewards (e.g., via exact answer matching) is often infeasible, so we adopt a cross-lingual semantic similarity signal as a practical proxy. Prior work, such as BlueBeri, shows that even surface-form similarity metrics (e.g., BLEU) can be competitive with learned reward models in single-language settings for such tasks. To further address your comment, we add a new experiment by evaluating different reward models on held-out preference pairs from CARE and HelpSteer3 (excluding the highest-ranked responses used as our references), and measure how well different reward functions align with human preferences. Our similarity-based reward achieves comparable overall accuracy to strong existing multilingual reward models (e.g., Skywork-Llama-3.1-8B), suggesting it provides a reasonable training signal. At the same time, our LRPO method could use any other reward models with the language router design; and we expect the performance could be even better if using better reward models.
>
> | Reward Model | Skywork-Llama-3.1-8B | Ours (140M parameters) |
> | ------------ | -------------------- | ---------------------- |
> | Accuracy (%) | 85.0                 | 84.4                   |
>
>
>
> We directly prompted gpt-oss-120b without any manual annotation. In practice, users can readily adapt LRPO by redefining the topic and region categories and using prompting or even fine-tuning a small LLM to tailor to their specific use cases.
>
> Our LRPO uses the same number of rollouts as GRPO, so the added overall computation time is limited to language routing and updating the router parameters, etc. We added a new computing-time analysis experiment inspired by your question. Interestingly, because our method encourages rollouts in non-English languages, where BPE tokenization typically produces 6% more tokens than English, responses tend to be longer, leading to a slight increase in inference time. For our experiment in Table 1, LRPO takes about 9.57 hours on our hardware 8 A40 GPUs, and GRPO takes 5.63 hours.
>
> |      | Rollout time per gradient step | Training time per gradient step |
> | ---- | ------------------------------ | ------------------------------- |
> | GRPO | 32.07 seconds                  | 327.24 seconds                  |
> | LRPO | 33.14 seconds                  | 538.57 seconds                  |
>
> >**W3/Q3: performances across model families**
>
> Performance differences across model families may stem from differences in pretrained multilingual capabilities, as also observed in prior work [1]. Nevertheless, within each model family, we compare against multiple post-training baselines and observe consistent improvements. To assess robustness, we conduct a paired bootstrap test at the language level (20,000 samples), consistent with our evaluation protocol. LRPO shows statistically significant improvements over GRPO on **Qwen (Seen: p=0.007, Avg: p=0.021)** and **Gemma (Avg: p=0.046)**, while gains on LLaMA are not statistically significant, possibly reflecting its weaker base multilingual capability. We will include these statistical significance results in the revision.
>
>
> >**Q2: reference responses in training data**
>
> Both datasets (HelpSteer & CARE) used in this paper are human preference datasets for studying RLHF algorithms (focus of our paper) that contain human ratings over multiple responses; we use the highest-rated response as the reference. Beyond this setup (out of scope of our paper), LRPO can be potentially paired with other reward signals, such as multilingual reward models or an LLM-as-a-judge. We will add a discussion in the paper to clarify this.
>
>
> [1] Y. Yue, Z. Chen, R. Lu, A. Zhao, Z. Wang, Y. Yue, S. Song, G. Huang. (2025). Does Reinforcement Learning Really Incentivize Reasoning Capacity in LLMs Beyond the Base Model? NeurIPS 2025.

---

> > ### Author Rebuttal · Reviewer_D52v · 2026-04-03
> >
> > Thank you for the author's responses. I believe the responses have added clarifications and addressed my concerns.
> >
> > I will adjust my overall score to 4, but note to the authors that I need to write the "Final Justification" section in order to be able to edit any fields (so the changes might not be there yet).

---

> > > ### Author Response · Authors · 2026-04-03
> > >
> > > Thank you again for providing your thoughtful review. If our responses have addressed your concern, we would appreciate it if you are willing to consider adjusting your score. We are also happy to answer any additional questions you may have.
> > >
> > > Thank you for considering **raising your score to 4**. We are glad that the additional experiments and clarification addressed your concerns. As we mentioned in the rebuttal, we will incorporate the updated discussion in the future version of the paper.

---

### Decision · Program_Chairs · 2026-04-30

**Decision:**

Accept (regular)

**Comment:**

The paper introduces Language-Routed Preference Optimization (LRPO), a framework for improving multilingual language model preference alignment by treating the response language as a selectable variable during preference optimization. Instead of generating rollouts only in the input language or relying on a fixed language, LRPO generates responses in multiple languages and learns to route training queries to the most informative languages using a trainable language router. Experiments across multiple model families (Qwen, LLaMA, Gemma) and multilingual benchmarks show that LRPO improves multilingual performance compared to prior preference optimization methods such as GRPO and other multilingual preference training approaches.

Reviewers are generally positive about the novelty of the proposed method but their are some issues raised on the weak performance on factual knowledge benchmarks (e.g. Reviewer GaXx) which was properly addressed by the authors.